# Adult Inception of Ketogenic Diet Therapy Increases Sleep during the Dark Cycle in C57BL/6J Wild Type and Fragile X Mice

**DOI:** 10.3390/ijms25126679

**Published:** 2024-06-18

**Authors:** Pamela R. Westmark, Timothy J. Swietlik, Ethan Runde, Brian Corsiga, Rachel Nissan, Brynne Boeck, Ricky Granger, Erica Jennings, Maya Nebbia, Andrew Thauwald, Greg Lyon, Rama K. Maganti, Cara J. Westmark

**Affiliations:** 1Department of Neurology, University of Wisconsin, Madison, WI 53706, USA; prwestmark@wisc.edu (P.R.W.); tswietlik@wisc.edu (T.J.S.); erunde2@wisc.edu (E.R.); bcorsiga@wisc.edu (B.C.); rnissan@wisc.edu (R.N.); bboeck@wisc.edu (B.B.); rhgranger@wisc.edu (R.G.); emjennings2@wisc.edu (E.J.); mnebbia@wisc.edu (M.N.); thauwald@wisc.edu (A.T.); gregory.lyon@focus.org (G.L.); maganti@neurology.wisc.edu (R.K.M.); 2Molecular Environmental Toxicology Center, University of Wisconsin, Madison, WI 53706, USA

**Keywords:** *Fmr1^KO^*, fragile X syndrome, ketogenic diet, electroencephalography (EEG), sleep

## Abstract

Sleep problems are a significant phenotype in children with fragile X syndrome. Our prior work assessed sleep–wake cycles in *Fmr1^KO^* male mice and wild type (WT) littermate controls in response to ketogenic diet therapy where mice were treated from weaning (postnatal day 18) through study completion (5–6 months of age). A potentially confounding issue with commencing treatment during an active period of growth is the significant reduction in weight gain in response to the ketogenic diet. The aim here was to employ sleep electroencephalography (EEG) to assess sleep–wake cycles in mice in response to the *Fmr1* genotype and a ketogenic diet, with treatment starting at postnatal day 95. EEG results were compared with prior sleep outcomes to determine if the later intervention was efficacious, as well as with published rest-activity patterns to determine if actigraphy is a viable surrogate for sleep EEG. The data replicated findings that *Fmr1^KO^* mice exhibit sleep–wake patterns similar to wild type littermates during the dark cycle when maintained on a control purified-ingredient diet but revealed a genotype-specific difference during hours 4–6 of the light cycle of the increased wake (decreased sleep and NREM) state in *Fmr1^KO^* mice. Treatment with a high-fat, low-carbohydrate ketogenic diet increased the percentage of NREM sleep in both wild type and *Fmr1^KO^* mice during the dark cycle. Differences in sleep microstructure (length of wake bouts) supported the altered sleep states in response to ketogenic diet. Commencing ketogenic diet treatment in adulthood resulted in a 15% (WT) and 8.6% (*Fmr1^KO^*) decrease in body weight after 28 days of treatment, but not the severe reduction in body weight associated with starting treatment at weaning. We conclude that the lack of evidence for improved sleep during the light cycle (mouse sleep time) in *Fmr1^KO^* mice in response to ketogenic diet therapy in two studies suggests that ketogenic diet may not be beneficial in treating sleep problems associated with fragile X and that actigraphy is not a reliable surrogate for sleep EEG in mice.

## 1. Introduction

Fragile X syndrome (FXS) is a genetic disorder caused by the loss of expression of fragile X messenger ribonucleoprotein (FMRP) and is comorbid with intellectual disability, autism, and seizures [1]. The most widely employed preclinical model for the study of FXS is the *Fmr1^KO^* mouse [2]; however, translating therapeutics from the mouse model to human clinical trials has been difficult. Identification of clinically relevant outcome measures that correlate between mice and humans is critical for FXS drug development [3]. Our prior work found hyperactivity in *Fmr1^KO^* mice during the first half of the light cycle when the mice should have been sleeping [4]. Almost half of children with FXS have sleep difficulties, including sleep onset insomnia, reduced total sleep time, and frequent awakenings [5,6,7,8,9,10,11,12]. Obstructive sleep apnea and nocturnal enuresis are also common [6,12,13,14,15]. Polysomnography indicates disrupted sleep microstructure, i.e., higher percentage of stage 1 non-rapid eye movement (NREM) sleep, lower percentage of rapid eye movement (REM) sleep, and increased REM latency in subjects with FXS compared to controls [16,17]. Considering the vitally important role sleep plays in memory consolidation, metabolism, mental health, cardiovascular function, and tissue growth and repair, therapies that restore normal sleep could greatly benefit FXS. Moreover, sleep electroencephalography (EEG) is a clinically translatable outcome measure.

Ketogenic diets (KDs) contain a high percentage of fat, moderate protein, and low carbohydrate and force the body into a state of ketosis that utilizes ketones instead of glucose as energy. KDs have been used for over a century to treat epilepsy, are currently a fad diet for weight loss, are showing improvements in core behavioral autism phenotypes, and rewire the circadian clock [18,19,20,21,22,23,24,25,26,27,28,29,30,31,32,33,34,35,36,37,38,39]. Regarding epilepsy, the KD decreases hyperexcitability of in vitro epileptic networks, improves diurnal rhythmicity in epileptic mice, and improves slow-wave sleep and sleep quality in children with refractory epilepsy [39,40,41,42]. Circadian rhythms control locomotor activity, feeding behavior, sleep–wake patterns, and other physiological and metabolic pathways [30,35]. The literature prompts testing the KD as a therapeutic strategy to improve sleep and ensuing sequelae in FXS.

The goals of this study were to compare sleep states in *Fmr1^KO^* and wild type (WT) littermate mice fed a KD commencing in adulthood versus a micronutrient-matched control diet and to compare the results with published work starting treatment at weaning [43]. EEG is the gold-standard clinical tool that quantitates sleep microstructure [44], but overnight polysomnography is not feasible in persons with severe developmental disabilities. This study also compares EEG findings with published actigraphy results [4]. Actigraphy is a less invasive, reduced-expense, indirect measure of rest–activity cycles with the potential to serve as a surrogate for EEG [45]. 

## 2. Results

### 2.1. Body Weight Is Reduced in Response to Ketogenic Diet

WT and *Fmr1^KO^* male littermate mice were randomly transferred to control and KD on postnatal day 95 (P95), maintained on respective diets throughout the study, and tested for EEG sleep phenotypes after 4 weeks of treatment, as described [43]. Similar to our prior studies [4,43], there was reduced body weight with KD, albeit not as severe as when commencing treatment at P18. Here, there was a 15% reduction in WT (*p* = 0.0011) and 8.6% reduction in *Fmr1^KO^* mouse (*p* = 0.0892) body weight after 4 weeks of treatment with KD (Appendix A). The prior study commencing treatment at P18 found a 54% (*p* < 0.0001) and 46% (*p* < 0.0001) decrease in WT and *Fmr1^KO^* mice, respectively, after 30 days of treatment [43]. Urine glucose and ketone levels were significantly elevated in *Fmr1^KO^* mice at 28 days KD treatment, but not statistically different in WT mice or in fasting blood at euthanasia (Appendix A).

### 2.2. Differences in 24 h Vigilance State Distribution Are Diet-Dependent

Days 4 and 6 out of 7 full days of EEG recordings were scored manually for sleep states by two blinded reviewers. Neither 24 or 12 h time bins indicated WT versus *Fmr1* genotype-specific effects in the percent of time spent awake, asleep, or in NREM or REM sleep in mice fed an AIN-76A purified-ingredient diet (Figure 1), which agrees with the prior study. KD initiated at P95 did not alter wake or sleep states as a function of the 24 h period in contrast to the prior study, where KD commenced at P18 decreased % wake and increased % sleep and % NREM sleep in both WT and *Fmr1^KO^* mice. 

### 2.3. KD Increases Time Spent in Sleep during the Dark Cycle

Mice exhibit higher activity levels during the dark cycle and increased sleep during the light cycle. Binning the EEG data by light and dark phases indicated that KD initiated at P95 decreased time spent in REM sleep in WT mice when the lights were on, which was observed in both WT and *Fmr1^KO^* mice in the prior study commencing KD at P18. When the lights were off, KD decreased time spent awake and increased time spent in sleep, NREM, and REM in both WT and *Fmr1^KO^* mice, concurring with the prior study (Figure 1). 

### 2.4. Sleep EEG Selectively Correlates with Actigraphy Findings during the Dark Cycle

Previously, we observed a genotype-specific difference in rest–activity patterns in adult male mice; specifically, *Fmr1^KO^* mice were hyperactive during the first half of the light cycle when nocturnal animals should be sleeping (Figure 3E in [4]). KD commencing at 2–3 months of age significantly reduced activity in *Fmr1^KO^* mice during the dark cycle when mice should be active. For direct comparison with the actigraphy data, EEG data were binned into 6 h increments (Appendix A). In contrast to actigraphy, there were no statistically significant genotype-specific differences in the EEG data binned in 6 h increments. In agreement with actigraphy, KD initiated at P95 decreased % wake (increased % sleep) during the dark cycle in *Fmr1^KO^* mice and also in WT. The increased % sleep during the dark cycle in response to KD can be attributed to increased % NREM during the first half of the dark cycle in both WT and *Fmr1^KO^* mice and to increased % NREM (WT) and % REM (WT and *Fmr1^KO^*) during the second half of the dark cycle. 

Binning the data into 2 h increments confirmed decreased wake time during the dark cycle in both WT and *Fmr1^KO^* mice fed KD initiated at P95, specifically during the beginning of the dark cycle and the 2 h before the light phase transition (Figure 2). There was a genotype-specific difference at bin 6 of the light cycle, with 37% increased wake in *Fmr1^KO^* compared to WT mice fed AIN-76A. The KD blunted both wake state peaks during the dark period, predominantly through increased NREM sleep. There were highly statistically different percentages for NREM and total sleep between the AIN-76A and KD cohorts during the dark cycle. Interestingly, in the prior study, there was a trend throughout the dark cycle in which KD increased % NREM in WT mice more than *Fmr1^KO^* [43], which was not observed here, suggesting that longer and/or earlier intervention with KD affects WT mice more than *Fmr1^KO^*. In contrast to the prior study, REM sleep was not significantly different during the light cycle as a function of KD initiated at P95, but both studies showed some increased REM during the dark cycle in response to KD in WT and *Fmr1^KO^* mice. Overall, treatment with KD flattened diurnal sleep periodicity curves in both wild type and *Fmr1^KO^* mice when initiated at P18 or P95, but a shorter maintenance period on chow versus AIN-76A here revealed a genotype-specific increase in NREM sleep during bin 6 of the light cycle in WT mice versus *Fmr1^KO^*. It remains to be determined how chow versus purified-ingredient diet affects sleep microstructure.

### 2.5. KD Increases the Number of NREM and REM Bouts during the Dark Cycle

Sleep microarchitecture analysis included the number and length of wake, sleep, NREM, and REM bouts over 24 h and 12 h bins. There were no genotype-specific differences in bout number (Figure 3), in contrast to the prior study where *Fmr1^KO^* mice treated with KD initiated at P18 exhibited a decreased number of wake, sleep, and NREM bouts over the 24 h period and during the dark cycle compared to WT treated with KD [43]. There were also no KD-induced differences in WT bout numbers over the 24 h bin or the 12 h light bin with dosing commencing in adults, whereas the prior study with earlier dosing indicated statistically significant increases in wake, sleep, and NREM bout number as a function of KD in WT mice over the 24 h period and during the 12 h dark cycle [43]. Both studies found an increased number of NREM bouts in WT and REM bouts in both WT and *Fmr1^KO^* mice in response to KD during the dark cycle. This study did not replicate increased sleep bout number in WT mice with KD during the light cycle or decreased REM bout number in *Fmr1^KO^* mice with KD during the light cycle.

### 2.6. KD Decreases the Length of Wake Bouts in Mice during the Dark Cycle

The average length of wake bouts during the 12 h dark cycle was significantly reduced in response to KD initiated at P95 in WT and *Fmr1^KO^* mice (Figure 4). The average reduction was 214 s for WT and 124 s for *Fmr1^KO^*. The reduced length of wake bouts in conjunction with a statistically equivalent number of wake bouts during the dark cycle indicates that the length of the wake bouts underlies reduced % wake time in response to KD during the dark phase. This study did not replicate prior findings of reduced average length of wake bouts over the 24 h period in WT mice in response to KD initiated at P18 or reduced average length of sleep bouts during the light cycle in WT mice in response to KD. There was a genotype-specific difference in average length of wake bouts with an average 126 s decrease in *Fmr1^KO^* compared to WT fed AIN-76A during the dark cycle here, but not in the prior study.

Analysis of the single longest wake bout over the 24 h period indicates a 64% decrease in bout length in WT mice in response to KD initiated at P95 (Figure 5), which is the same as the prior study. Here, we did not replicate the finding in *Fmr1^KO^* mice, nor the decrease in the longest REM bout in WT mice in response to KD over the 24 h period. However, we did observe a genotype-specific difference in longest wake bout in *Fmr1^KO^* versus WT fed AIN-76A over the 24 h period (34% decrease), as well as a 35% decrease during the dark cycle. The KD significantly decreased the longest wake bout in both WT and *Fmr1^KO^* mice during the dark cycle in both studies. There was no difference in the latency time to the first sleep bout as a function of genotype or diet (Appendix A).

During REM scoring, the EEG pattern suggested the mice were trying to enter REM but failing. The EEG pattern needed to exhibit seven or more 4 s epochs of low-amplitude/high-frequency waves to be scored as REM. Instances of one to six epochs of low-amplitude/high-frequency waves were tallied as failed attempts to enter REM. The prior study indicated a statistically significant 20% decrease in failed REM attempts in *Fmr1^KO^* mice compared to WT fed AIN-76A during the dark cycle and was the only genotype-specific EEG phenotype in the mice dosed with the control diet since weaning. Here, in mice commencing dosing as adults, we found a 12% decrease in failed REM attempts in *Fmr1^KO^* mice compared to WT fed AIN-76A during the dark cycle. We also observed statistically significant decreases over the 24 h period and during the dark cycle comparing WT and *Fmr1^KO^* mice fed KD initiated at P95 (Figure 6).

### 2.7. Fmr1^KO^ Mice Do Not Exhibit Epileptic Activity on Sleep EEG

Sleep activates centrotemporal discharges characteristic of focal motor seizures in FXS [46,47]. Similar to Pietropaolo and colleagues [48], as well as our prior EEG study [43], we did not observe seizures, absences, or interictal events in any WT or *Fmr1^KO^* mouse in scored EEG recordings.

## 3. Discussion

The validation of outcome measures that translate between preclinical and clinical studies is imperative for the FXS field [49]. Our laboratory is interested in validating actigraphy and polysomnography as outcome measures that translate between mice and humans and in identifying dietary and pharmaceutical interventions for FXS. KD therapy has shown success in treating epilepsy, repetitive behavior, intellectual impairment, language dysfunction, and social skills, as well as in improving sleep, i.e., increasing REM sleep and reducing nighttime awakenings and daytime sleepiness [50,51,52]. Previously, we quantitated EEG sleep metrics in adult mice commencing KD therapy at weaning at P18 and did not find good correlation between sleep as assessed by EEG and activity levels assessed by actigraphy [4,43]. The large deficit in body weight in response to KD during development could have been a confounding factor. Here, we quantitated sleep states by EEG in adult *Fmr1^KO^* and littermate mice in response to KD therapy commenced in adulthood (P95). 

The major difference we observed between the two sleep studies was a genotype-specific difference in sleep during the middle of the light cycle in mice on the control diet initiated at P95. Specifically, in this study, WT mice slept more than *Fmr1^KO^* mice, which concurs with the human phenotype. The control diet in both studies was AIN-76A, which was matched for vitamin and mineral content with the KD; however, mice were transferred from the standard vivarium chow (2019 Teklad Global 19% Extruded Rodent Diet) to AIN-76A or KD at P18 in the prior study [43] and P95 here. The longer maintenance time on the chow could account for the altered sleep phenotype, as purified-ingredient diets have been shown to reduce seizure propensity, body weight, and lean mass in mice compared to chows [53,54,55], which could mask genotype differences. Reduced sleep in *Fmr1^KO^* mice compared to WT on the control diet during the middle of the light cycle overlaps with published actigraphy data in mice undergoing a similar KD treatment protocol, where *Fmr1^KO^* mice exhibit a 38% increase in activity during the first half of the light cycle (Figure 3E in [4]). Selectively increased hyperactivity in *Fmr1^KO^* mice during the first half of the light cycle concurs with the sleep phenotype in the human disorder, where children with FXS have trouble falling asleep. Improved sleep in WT mice during the light cycle is likely due to a combination of increased NREM (bin 6, 15% over *Fmr1^KO^*, *p* = 0.0516) and REM sleep (bin 6, 13% increase over *Fmr1^KO^*, *p* = 0.4068).

Percent sleep assessed by EEG during the middle of the light cycle was reduced in both KD cohorts compared to WT while being similar to *Fmr1^KO^* mice on the control diet. The overlap between KD cohorts and the *Fmr1^KO^*/AIN-76A cohort during the typical mouse sleep period indicates that the KD is not therapeutic in rescuing sleep in the *Fmr1^KO^* mice but rather causes a sleep deficit in WT mice. KD therapy was highly effective at reducing audiogenic-induced seizures (AGSs) in *Fmr1^KO^* mice, which were tested during the light cycle [4]. In total, these data suggest that sleep deprivation does not underlie seizure propensity in the *Fmr1^KO^* mouse, as KD therapy can correct the AGS phenotype but not the sleep deficit. 

In both studies, we found that KD increased NREM and REM sleep during the dark cycle in WT and *Fmr1^KO^* mice, which corresponds to reduced activity levels in *Fmr1^KO^* commencing treatment at postnatal or adult ages as well as with reduced activity levels in WT mice treated with KD at P18, but not with WT mice commencing treatment at P95 where there is statistically equivalent activity levels with/without KD [4,43]. Overall, actigraphy does not appear to be a reliable surrogate for sleep EEG. Increased sleep in mice during lights off when mice are supposed to be active is suggestive of somnogenic effects of the KD. In contrast, excessive daytime sleepiness in humans improves with KD therapy [56].

There have been numerous recent studies, in human and rodent models of FXS, identifying EEG biomarkers [57,58,59,60,61,62,63,64,65,66,67,68,69,70,71,72,73,74,75,76,77]. To our knowledge, our study was the first published work conducting 24 h polysomnography recordings in *Fmr1^KO^* mice [43]. A preprint by Martinez and coworkers also conducted 24 h polysomnography recordings in adult *Fmr1^KO^* mice in the C57BL/6J background and found reduced NREM sleep in *Fmr1^KO^* [78].

The limitations of this study include only testing adult animals and only testing males. Larger sleep deficits may be observed in juvenile-age animals. We only tested EEG in adult mice because the weight of the headcaps in combination with thinner skull bones in rapidly growing adolescent-age animals would have presented insurmountable technical challenges with our current system. We only tested one sex due to the labor-intensive nature of manual scoring of EEG recordings. We tested males because WT and *Fmr1^KO^* littermates could be generated from *Fmr1^HET^* dams and because it is expected that there are larger differences between WT and *Fmr1^KO^* males than between *Fmr1^HET^* and *Fmr1^KO^* females. Previously, we found that KD selectively attenuates seizures, reduces body weight, and decreases activity levels in male but not female *Fmr1^KO^* mice [4]. This study overcomes a major limitation of our prior work [43] by commencing KD treatment in adult animals and circumventing the extreme reduction in body weight in mice maintained on KD from weaning.

## 4. Conclusions

In summary, we tested the efficacy of KD on sleep EEG in *Fmr1^KO^* mice because ketogenic high-fat diets are proving therapeutic for many diseases that are comorbid with FXS, including epilepsy and autism, and persons with FXS present with an abnormal fatty acid profile [79]. Our prior work found that KD was as or more effective than the best metabotropic glutamate receptor 5 (mGluR_5_) inhibitors in reducing seizures in the AGS assay. Here, we found a sleep deficit in *Fmr1^KO^* mice, but we did not observe therapeutic efficacy with regard to KD and sleep outcomes. Rather, KD induces an *Fmr1^KO^* sleep phenotype in WT mice. The novelty of the current study commencing KD at P95, versus the prior study initiating KD at P18, is the genotype-specific NREM sleep deficit in *Fmr1^KO^* mice during bin 6 of the light cycle. Overall, data from the two studies suggest that actigraphy is not a reliable surrogate for sleep EEG in mice. The lack of evidence for improved sleep during the light cycle in *Fmr1^KO^* mice suggests that the KD may not be beneficial in treating sleep problems associated with the disorder, although there are significant effects on seizures and activity levels. It remains to be determined how sleep EEG responds to pharmaceutical interventions, such as mGluR_5_ inhibitors, and to exposome factors in mouse models of FXS. Studies are underway to test sleep outcomes in response to purified-ingredient versus chow diets.

## 5. Materials and Methods

### 5.1. Materials and Mice

Materials and supplier information have been published previously [43]. The *Fmr1^tm4Cgr^* (*Fmr1^KO^*) mice were originally developed by the Dutch-Belgian FXS Consortium and backcrossed > 11 times to FVB mice [2]. They were backcrossed into the C57BL/6J background by Dr. Bill Greenough’s laboratory (University of Illinois at Urbana-Champaign) and distributed to other laboratories. We have maintained the *Fmr1^KO^* mice in the C57BL/6J background at the University of Wisconsin-Madison for over 20 years with occasional backcrossing with C57BL/6J mice from Jackson Laboratories to avoid genetic drift. Breeding pairs for these experiments were housed in static microisolator cages with ad libitum access to food (2019 Teklad Global 19% Extruded Rodent Diet) and water in a temperature- and humidity-controlled vivarium on a 12 h light cycle. The bedding (Shepherd’s Cob + Plus, ¼ inch cob) contained nesting material as the only source of environmental enrichment. All animal husbandry, surgery, and euthanasia procedures were performed under NIH and an approved University of Wisconsin-Madison animal care protocol administered through the Research Animal Resources Center with oversight from the Institutional Animal Care and Use Committee (IACUC). *Fmr1* genotypes were determined by PCR analysis of DNA extracted from tail biopsies with HotStarTaq polymerase (Qiagen Inc., Germantown, MD, USA; catalog #203205) and Jackson Laboratories’ (Bar Harbor, ME, USA) primer sequences oIMR2060 [mutant forward; 5′- CAC GAG ACT AGT GAG ACG TG-3′], oIMR6734 [WT forward; 5′-TGT GAT AGA ATA TGC AGC ATG TGA-3′], and oIMR6735 [common reverse; 5′-CTT CTG GCA CCT CCA GCT T-3′], which produced PCR products of 400 base pairs (*Fmr1^KO^*) and 131 base pairs (WT). Heterozygote females exhibited both the 400 and 131 base pair bands. *Fmr1^HET^* females were bred with *Fmr1^KO^* males and generated WT and *Fmr1^KO^* male littermate mice for the described experiments. Experimental animals were derived from multiple breeding pairs, as recommended [80,81]. Mice were randomly weaned onto control and KDs at postnatal P95, maintained on their respective diets throughout the rest of the study, and tested for EEG sleep phenotypes after 28 days of treatment. Mice were socially housed prior to surgery and individually housed after EEG electrode implantation. Cohorts included WT-fed AIN-76A (*n* = 12), WT-fed KD (*n* = 10), *Fmr1^KO^*-fed AIN-76A (*n* = 11), and *Fmr1^KO^*-fed KD (*n* = 12).

### 5.2. EEG Electrode Implantation

Right frontal, left parietal, and occipital electrodes were placed along with two stainless steel wire electrodes in nuchal muscles for electromyography (EMG) under isoflurane anesthesia according to the methods described previously [43,82]. The occipital electrode served as the ground and as the reference electrode for the surgical recording procedure. Mice were allowed to recover from surgery for 3 days prior to transfer to tethered EEG recording setups, with mice housed individually in Plexiglas^®^ chambers for sleep recording. EEG signals were acquired for 7 days (across the light and dark cycles) on an XLTEK machine (Natus, Madison, WI, USA) with a 512 Hz sampling rate. EDF Browser software (v.2.40) was used to convert Natus EEG recordings into files compatible with Sirenia^®^ Sleep Pro software v.1.3.2 (Pinnacle Technology, Lawrence, KS, USA). EEG signals from days 4 and 6 were scored by hand with Sirenia^®^ Sleep Pro software in 4 s epochs for wake, REM, and NREM states by two scorers blinded with respect to treatment group, as previously described [43,82,83,84]. Waking epochs were identified as those with high EMG amplitude and epochs with little EMG activity were scored as sleep. Specific sleep states were scored based on predominant EEG power, where NREM was associated with low-frequency and high-amplitude delta (1–4 Hz) activity and REM was associated with increased-frequency but low-amplitude theta (5–7 Hz) activity. Data were binned into 2, 6, 12, and 24 h increments for analyses. Hypnograms for all mice are provided in the Appendix A.

### 5.3. Biometrics

Ketone and glucose levels were assessed in urine or blood as indicated in the figure legend using a Precision Xtra blood glucose and ketone monitoring system (Abbott Diabetes Care Inc., Alameda, CA, USA). For urine samples, mice were gently restrained at the neck, and urine was collected. For blood samples, mice were anesthetized with isoflurane and blood collected from the inferior vena cava and assayed for ketone and glucose levels with a Precision Xtra meter. Low off-scale glucose meter readings were adjusted to 20 mg/dL. High off-scale glucose meter readings were adjusted to 500 mg/dL. High off-scale ketone meter readings were adjusted to 8.0 mmol/L. Blood samples were collected during the light phase after 4 h of fasting.

### 5.4. Statistical Analysis

Data were analyzed with Microsoft^®^ 365 Excel for Mac version 16.66.1, Microsoft Office Professional Plus 2019 version 10.2.1 (339), and GraphPad Prism version 10.2.1 (339) software. The vast majority of the data passed the D’Agostino and Pearson test for normal Gaussian distribution, and all data were analyzed similarly with parametric ANOVA tests. Average data were presented ±SEM. Statistical significance was determined by 2-way ANOVA with Tukey’s multiple comparison tests for Figure 1, Figure 3, Figure 4, Figure 5 and Figure 6 and 2-way ANOVA with repeated measures and post-hoc Tukey multiple comparison tests for Figure 2. 

## Figures and Tables

**Figure 1 ijms-25-06679-f001:**
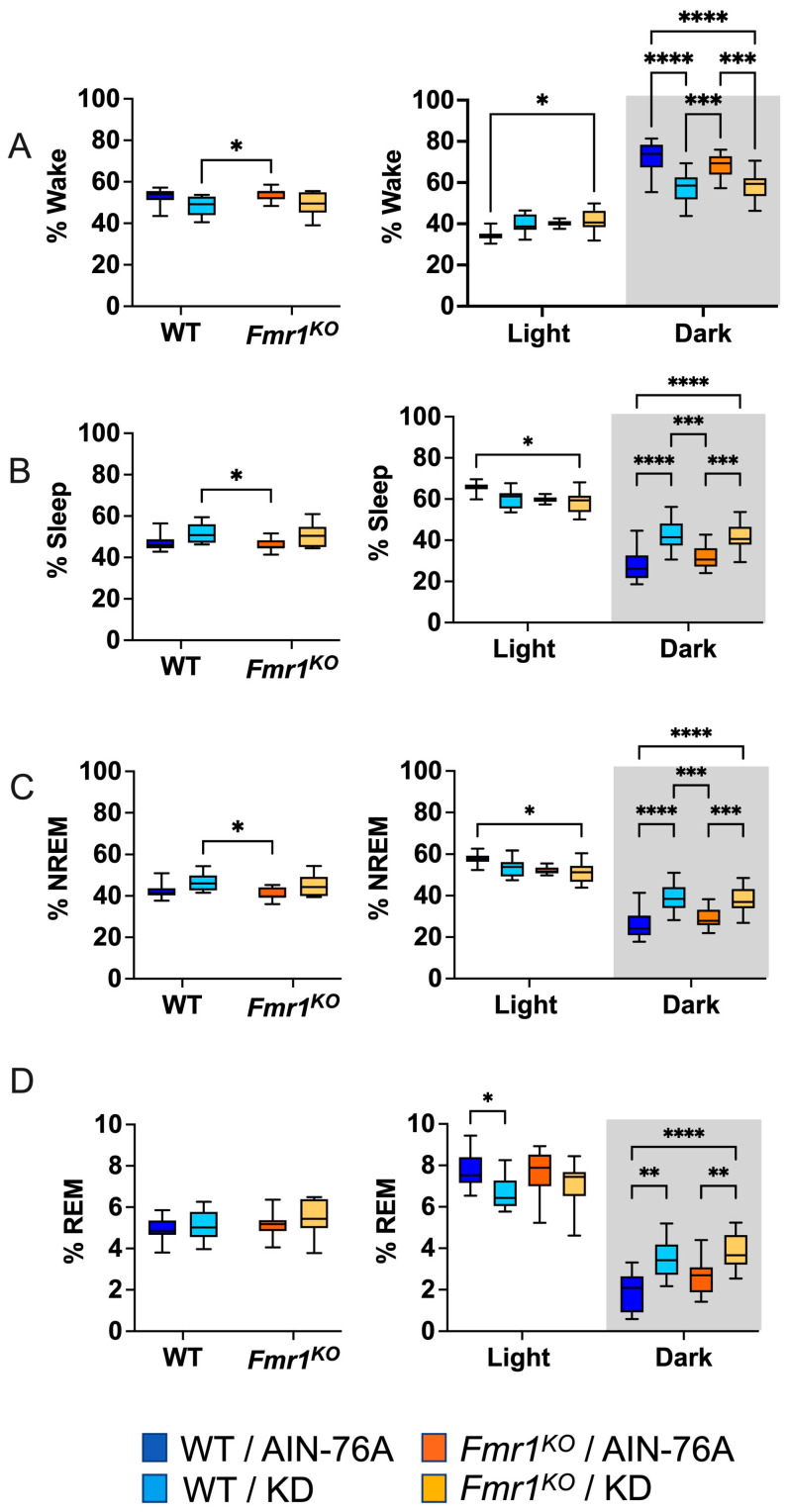
Effect of KD initiated at P95 on sleep architecture in WT and *Fmr1^KO^* mice during 24 h cycle and as a function of light and dark cycles. EEG/EMG recordings were scored for wake, sleep, NREM, and REM activity, and the data were parsed into 24 h full day and 12 h light/dark bins starting at Zeitgeber time zero. The average percent sleep state was plotted versus bin for (**A**) % wake, (**B**) % sleep, (**C**) % NREM, and (**D**) % REM. Key for statistical significance: * *p* ≤ 0.05, ** *p* < 0.01, *** *p* < 0.001, **** *p* < 0.0001, etc.

**Figure 2 ijms-25-06679-f002:**
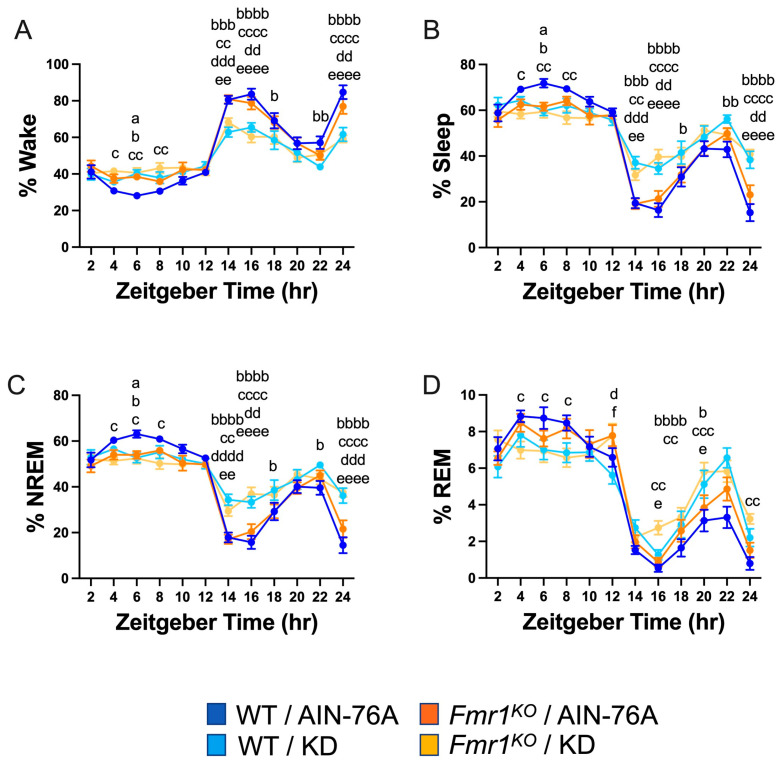
Effect of KD initiated at P95 on sleep architecture in WT and *Fmr1^KO^* mice as a function of time of day. EEG/EMG recordings were scored for wake, sleep, NREM, and REM activity, and the data were parsed into 2 h bins starting at Z0 (lights on). The average percent sleep state was plotted versus 2 h time bins for (**A**) % wake, (**B**) % sleep, (**C**) % NREM, and (**D**) % REM. Key for statistical significance: WT/AIN-76A versus *Fmr1^KO^*/AIN-76A = “a”, WT/AIN-76A versus WT/KD = “b”, WT/AIN-76A versus *Fmr1^KO^*/KD = “c”, *Fmr1^KO^*/AIN-76A versus WT/KD = “d”, *Fmr1^KO^*/AIN-76A versus *Fmr1^KO^*/KD = “e”, and WT/KD versus *Fmr1^KO^*/KD = “f”; a *p* ≤ 0.05, b *p* ≤ 0.05, bb *p* < 0.01, bbb *p* < 0.001, bbbb *p* < 0.0001, c *p* ≤ 0.05, cc *p* < 0.01, ccc *p* < 0.001, cccc *p* < 0.0001, d *p* ≤ 0.05, dd *p* < 0.01, ddd *p* < 0.001, dddd *p* < 0.0001, e *p* ≤ 0.05, ee *p* < 0.01, eeee *p* < 0.0001, f *p* ≤ 0.05.

**Figure 3 ijms-25-06679-f003:**
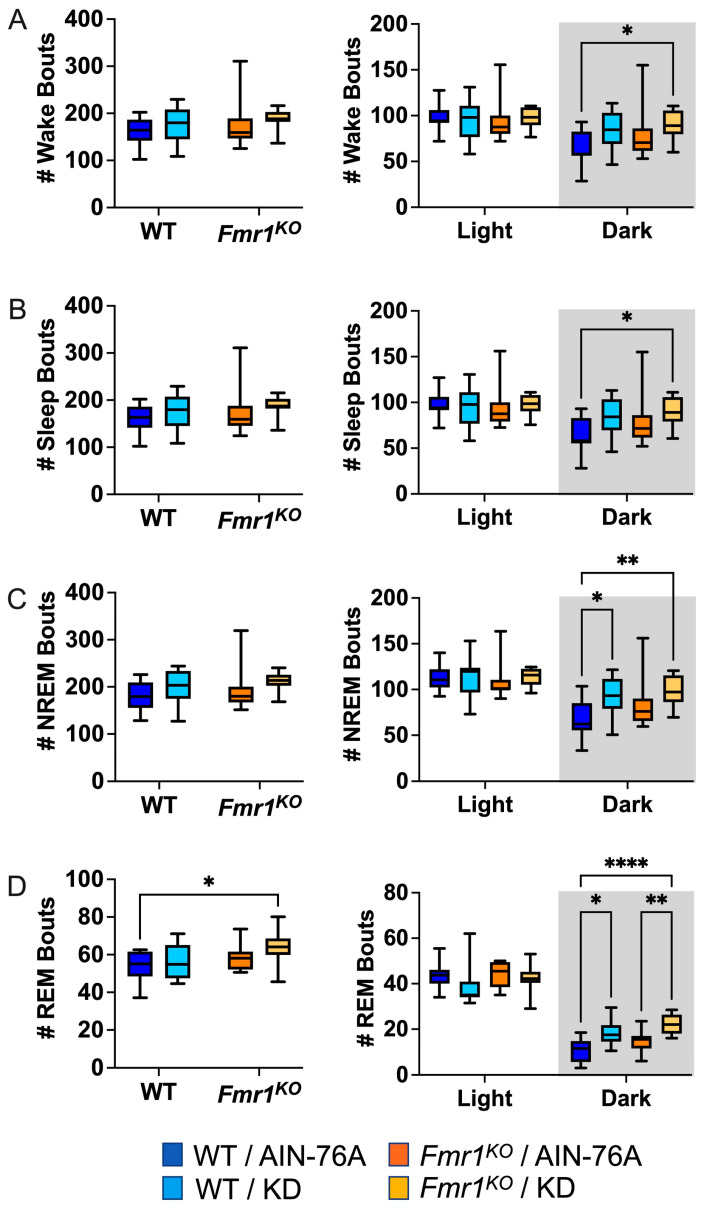
Effect of KD initiated at P95 on sleep microarchitecture (number of bouts) in WT and *Fmr1^KO^* mice as a function of light and dark cycles. EEG/EMG recordings were scored for wake, sleep, NREM, and REM activity, and the data were parsed into 24 h full day and 12 h light/dark bins starting at Z0. The average number (#) of sleep/wake state bouts was plotted versus bin for (**A**) wake bouts, (**B**) sleep bouts, (**C**) NREM bouts, and (**D**) REM bouts. Key for statistical significance: * *p* ≤ 0.05, ** *p* < 0.01, **** *p* < 0.0001, etc.

**Figure 4 ijms-25-06679-f004:**
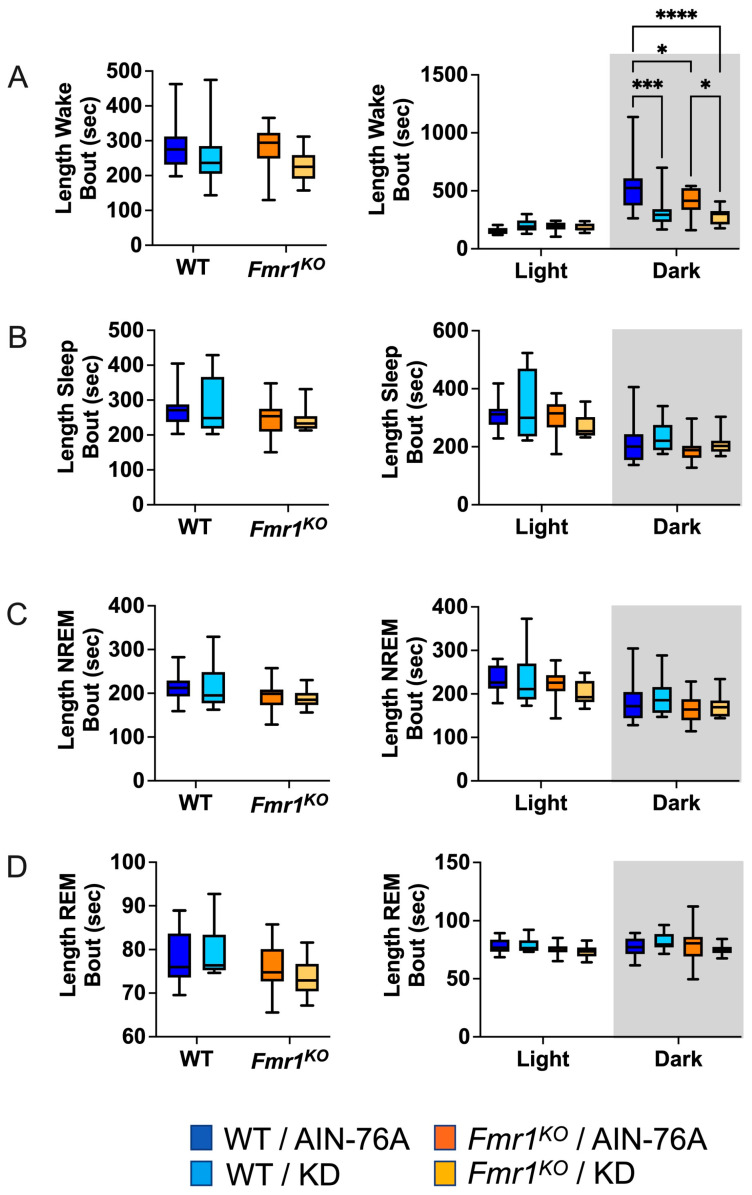
Effect of KD initiated at P95 on sleep microarchitecture (length of bouts) in WT and *Fmr1^KO^* mice as a function of light and dark cycles. EEG/EMG recordings were scored for wake, sleep, NREM, and REM activity, and the data were parsed into 24 h full day and 12 h light/dark bins starting at Z0. The average length of time of sleep/wake bouts was plotted versus bin for (**A**) wake bouts, (**B**) sleep bouts, (**C**) NREM bouts, and (**D**) REM bouts. Key for statistical significance: * *p* ≤ 0.05, *** *p* < 0.001, **** *p* < 0.0001, etc.

**Figure 5 ijms-25-06679-f005:**
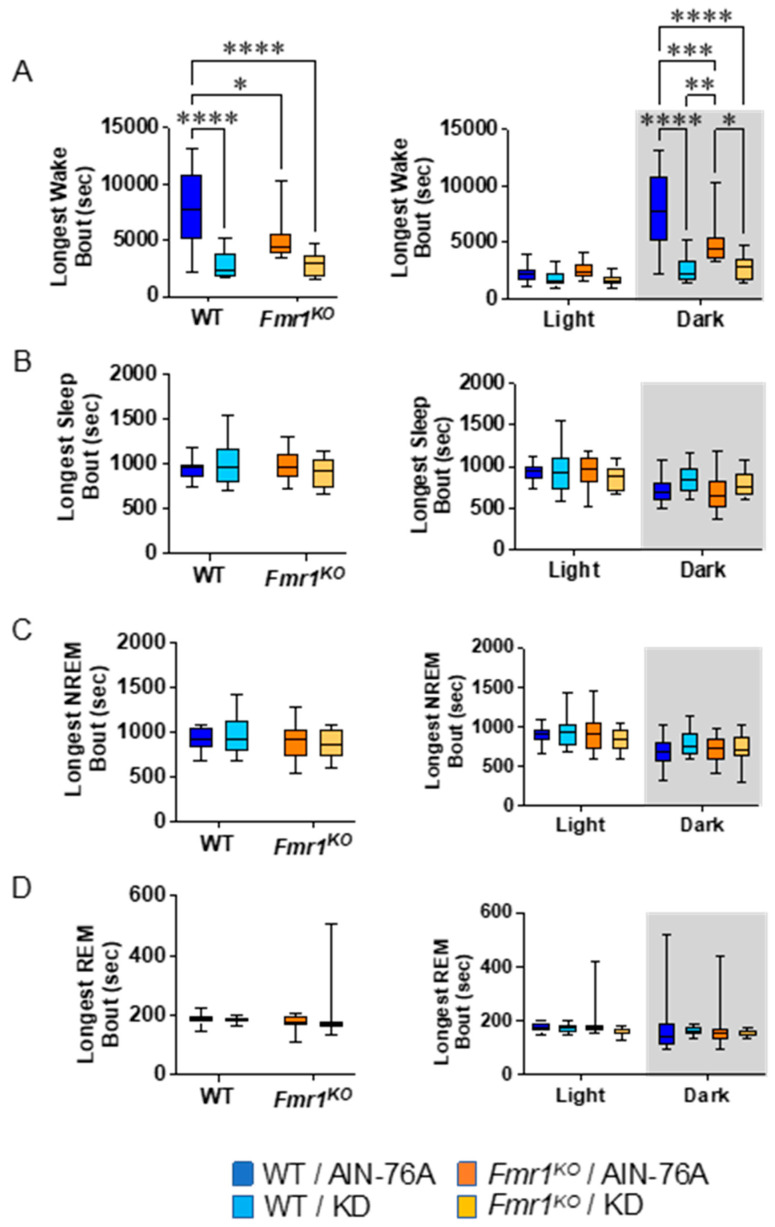
Effect of KD initiated at P95 on additional sleep microarchitecture phenotypes in WT and *Fmr1^KO^* mice as a function of light and dark cycles. EEG/EMG recordings were scored for wake, NREM, and REM activity, and the data were parsed into 24 h full day and 12 h light/dark bins starting at Zeitgeber time zero. Phenotype averages were plotted versus bin for (**A**) longest wake bout, (**B**) longest sleep bout, (**C**) longest NREM bout, and (**D**) longest REM bout. Key for statistical significance: * *p* ≤ 0.05, ** *p* < 0.01, *** *p* < 0.001, **** *p* < 0.0001, etc.

**Figure 6 ijms-25-06679-f006:**
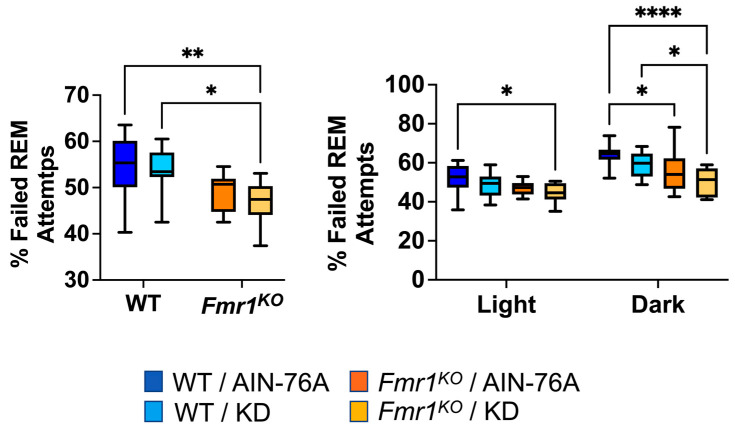
Effect of ketogenic diet initiated at P95 on ability to enter REM sleep in WT and *Fmr1^KO^* mice. EEG/EMG recordings were scored for the number of attempts to enter REM and expressed as a percentage of total attempts. Key for statistical significance: * *p* ≤ 0.05, ** *p* < 0.01, **** *p* < 0.0001, etc.

## Data Availability

Data are contained within the article or Appendix A. Sirenia EEG/EMG files are available from the corresponding author upon request.

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
