# Peer review of "Adult Inception of Ketogenic Diet Therapy Increases Sleep during the Dark Cycle in C57BL/6J Wild Type and Fragile X Mice"

_ijms, 2024, doi:10.3390/ijms25126679_

Round 1

Reviewer 1 Report

Comments and Suggestions for Authors

The article entitled  Ketogenic Diet Affects Sleep Architecture in C57BL/6J Wild Type and Fragile X Mice, Part 2.0 in which the authors assessed sleep-wake cycles in Fmr1KO male mice and wild type (WT) liAermate controls in response to ketogenic diet therapy where mice were treated from weaning (postnatal day 18) through study completion (5-6 months of age).

The manuscript is based on a good idea and is worth of publication but after some revisions.

The authors should revise their manuscript according to the following comments.

Abstract

Should be restructured as this one is confusing. The authors should follow the following format.

Background, Aim, Methods, conclusion,

Introduction

Line 61-71

ketogenic diet (KD) versus a micronutrient-matched control diet.

The authors should add more detail or an extra paragraph here as diet is linked with other inflammatory diseases as well.

The authors can read and cite the following articles which are recently published in 2024 and are very interesting which I randomly found on Researchgate,

1)    Elucidating the Role of Diet in maintaining the Gut Health to Reduce the Risk of Obesity, Cardiovascular and other age-related Inflammatory Diseases: Recent Challenges and Future Recommendations.

2)    Dietary Implications of the Bidirectional Relationship between the Gut Microflora and Inflammatory Diseases with special emphasis on Irritable Bowel Disease: Current and Future Perspective.

Results are well elaborated.

Figures’ quality is good.

The methodology is well-designed.

Discussion is fine.

Conclusion should be added.

Author Response

We thank the reviewers for their positive and careful critique of our work. Please find responses to their queries attached.

Reviewer 2 Report

Comments and Suggestions for Authors

General comment: In many parts, text refers to previous published results, mainly in the Result section, and that affects the Discussion section, which may be extended with some sentences that are mentioned in the results. In the end of the paper, the last paragraph seems like you are summarizing the whole investigation (parts 1 and 2), and the beginning of it looks like a part of an abstract. Suggestion on adding a couple of sentences where you briefly conclude novelties from current results.

Minor revision

Suggestion on renaming the title of the paper so that it emphasizes what is new in this research in comparison to previous research and avoids the use of the same title and Part 2. That should also be emphasized in the aim and whole text of the results. It seems like the conceptualization of the introduction is like a previous paper published in the same journal.

 The affiliation should be rewritten without repetition of the Department of Neurology, University of Wisconsin, Madison.

Define the aim of the study precisely. In the end of the introduction, the purposes of the study are described very well, but without a precise definition of the aims.

 Subtitle 2.1. Study Design seems inappropriate for the Results section, suggesting renaming it to, for example, Body weight reduction.

 Different panels of figures are not marked in accordance with instructions for authors of the journal (https://www.mdpi.com/journal/ijms/instructions#figures); for example, (A) should be replaced with (a) in proper font.

 Citation Figure 3E in [4] in line 123 is repeated in this paragraph.

 Lines 154-157 in Figure 2 description should rather be in the method section. Same for lines 180-182, 220-224, 231-233, 241-242.

 In line 160 (and a p<0.05, aa p<0.01, aaa p<0.001, aaaa p<0.0001) suggesting adding etc. to emphasize that the same applies for b, c…

The font size in figures is not consistent. For example, the font size in the legend of Fig. 5 is much bigger than the rest.

Lines 254-258 contain some parts that are already mentioned in the introduction.

In lines 270 and 342 use the hole name 2019 Teklad Global 19% Protein Extruded Rodent Diets rather than just Teklad 2019 to avoid misunderstandings for people who are not so familiar with the field.

Line 331 starts with:

 Materials

Materials and supplier information has been published previously [43].

Comment: Recommendation to say something about materials from [43] or add this citation in 4.2. part.

 Line 399: The number of subjects, genotype and treatment details are included in the figure legends.

Comment: Recommendation to write about statistical tests in this part rather than in figure explanations.

Author Response

(The authors gave the same response as above.)
